# Total, Added, and Free Sugar Consumption and Adherence to Guidelines in Switzerland: Results from the First National Nutrition Survey menuCH

**DOI:** 10.3390/nu11051117

**Published:** 2019-05-19

**Authors:** Angeline Chatelan, Pierre Gaillard, Maaike Kruseman, Amelie Keller

**Affiliations:** 1Center of Primary Care and Public Health (Unisanté), University of Lausanne, Route de la Corniche 10, 1010 Lausanne, Switzerland; pierre.gaillard@etu.hesge.ch; 2School of Health Sciences, HES-SO University of Applied Sciences and Arts Western Switzerland, Rue des Caroubiers 25, 1227 Carouge, Geneva, Switzerland; maaike.kruseman@hesge.ch; 3Research Unit for Dietary Studies at the Parker Institute, Bispebjerg og Frederiksberg Hospital, The Capital Region, Nordre Fasanvej 57, 2000 Frederiksberg, Denmark; amelie.cleo.keller@regionh.dk

**Keywords:** sugar intake, dietary survey

## Abstract

The World Health Organization (WHO) recommends reducing free sugars to less than 10% of total energy intake (TEI) due to their potential implications in weight gain and dental caries. Our objectives were to (1) estimate the intake of total, added, and free sugars, (2) define the main sugar sources, and (3) evaluate the adherence to sugar guidelines. The first national nutrition survey 2014–2015 included non-institutional adults aged 18–75 years. Diet was assessed with two non-consecutive 24-hour dietary recalls in 2057 participants. Added and free sugar content was systematically estimated by two dietitians using available information from the manufacturer and/or standard recipe/composition. Usual daily intake distributions were modeled and weighted for sampling design, non-response, weekdays, and seasons. Total, added, and free sugar intake was respectively 107 g (±44), 53 g (±36), and 65 g (±40), representing 19%, 9%, and 11% of TEI. Sugar consumption was higher among younger adults and lower among people living in the Italian-speaking region. The three main food sources of free sugars were: (1) sweet products (47% of total free sugars), in particular sweet spreads (15%) and cakes/cookies (11%); (2) beverages (29%), mainly fruit and vegetable juices (13%), and sugar-sweetened beverages (12%, but 20% in younger adults); and (3) dairy products (9%), with yogurt accounting for 6%. Respectively, 44% of women and 45% of men had free sugar intake below 10% of TEI. Of people aged between 18–29, 30–64, and 65–75 years, 36%, 45%, and 53% had free sugar intake below 10% of TEI, respectively. The prevalence of Swiss people with free sugar intake that was <5% of the TEI was 8%. Adherence to the WHO recommendations guidelines was generally low in Switzerland, particularly among young adults, and in line with other high-income countries.

## 1. Introduction

Several systematic reviews and meta-analyses have concluded that an increased intake of sugar and sugar-sweetened beverages (SSBs) is directly associated with weight gain, overweight, and obesity [1,2,3]. The intake of dietary sugars has also a deleterious effect on oral health, and is the most important risk factor for dental caries [4,5]. In addition to dental caries and body weight, excessive sugar consumption has been associated with an increased risk of developing several chronic diseases, such as type 2 diabetes [6], cardiovascular diseases [7,8,9], and some cancers [10,11,12], as well as non-alcoholic fatty liver disease [13]. Emerging research suggests that a diet high in sugars may increase the risk of developing dementia such as Alzheimer disease [14,15]. In this context, the World Health Organization (WHO) recommends a reduced intake of free sugars throughout the life course with a reduction of free sugars intake to less than 10% of total energy intake (TEI, strong recommendation) and preferably below 5% of TEI (conditional recommendation) in both adults and children [16]. In Switzerland—a wealthy country with three main linguistic regions, one of the highest life expectancies [17], and the lowest prevalence of obesity in high-income countries [18]—the maximum daily intake for free sugars is set at 50 g, corresponding to 10% of a TEI equivalent to 2000 kcal [19]. 

In Europe, sugar consumption currently contributes between 7–11% and 11–17% of TEI in adults and children, respectively [20]. Worldwide, data suggest that added sugar intake rises starting from one year of age, and is highest among school-age children and adolescents compared to adults [21]. Sweet products (i.e., cakes, biscuits, pastries, confectionary, jam/honey, ice cream, table sugar), beverages (i.e., SSBs and fruit nectars, excluding fruit juices) and dairy products (i.e., yoghurts, milk-based desserts) contribute the most to added sugar intake in both adults and children [20,22]. 

Around the world, a variety of public health policies have been designed to limit the intake of added or free sugar, such as food labeling (i.e., Nutri-score) [23], food reformulation [24], a reduction of portion sizes [25], and the taxation of SSBs [26]. In Switzerland, several policy proposals or public health interventions are being considered [27]. However, in order for these policies to be evaluated in the future, reliable data about the current sugar intake of the population is needed. These data are also necessary for monitoring purposes and to investigate who are the most at risk of sugar overconsumption. Yet, in Switzerland, the population’s sugar consumption has not previously been reported. Hence, the aims of this study were (1) to estimate the intake of total, added, and free sugars, (2) to define the main sugar sources at the food group level, and (3) to evaluate the adherence to sugar guidelines in the Swiss adult population using data from the first national nutrition survey, menuCH. 

## 2. Methods

### 2.1. Study Design and Population

The cross-sectional Swiss National Nutrition Survey, menuCH, was conducted among non-institutionalized residents aged 18–75 years old [28]. Participants were recruited from the national sampling frame for person and household surveys [29] using a stratified random sample design. The survey population was intended to be representative of the Swiss population in terms of age and place of residency (all seven major areas of Switzerland) [28]. A total of 5496 eligible people reachable by phone were invited, of which 2086 (38%) responded [28]. Participants and non-participants had similar age and marital status, but participants were more frequently women and Swiss nationals [28]. Data were collected from January 2014 to February 2015. Further information about menuCH is available here: https://menuch.iumsp.ch. The study was registered in the trial registry (identification number: ISRCTN16778734) and conducted according to the guidelines laid down in the Declaration of Helsinki. Each participant signed a written informed consent. 

### 2.2. Dietary Assessment

Trained dietitians conducted two non-consecutive 24-h recalls (24HDR, first: face-to-face and second: by phone, two to six weeks later). They used the computer-directed interview program GloboDiet^®^, which was previously known as EPIC-Soft^®^ (International Agency for Research on Cancer (IARC), Lyon, France) [30,31] following standardized steps for 24HDR: (1) general participant information (e.g., special diet, special day); (2) quick list of food consumption occasions and items; and (3) detailed description and quantification of all consumed foods and beverages, including conservation and preparation methods, sugar content/addition, and portion size. To support survey participants in quantifying consumed amounts, dietitians used a book with 119 series of six graduated portion-size pictures [32]. 24HDR were spread over all weekdays and seasons. According to the description reported by the dietitians, each food item was linked to the best match in an extended research version of the 2015 Swiss Food Composition Database [33] using FoodCASE (Premotec GmbH, Winterthur, Switzerland) [34]. We classified all foods and beverages into 30 food groups, based on their original classification from GloboDiet^®^ (18 groups and 85 subgroups) and on their nutritional similarity regarding sugar content.

### 2.3. Total, Added, and Free Sugars: Definitions, Calculations, and Recommendations

Appendix A provide an overview of the total, added, and free sugar definitions. *Total sugars* correspond to all monosaccharides and disaccharides present in food and derived from any source [16,35]. The Swiss Food Composition Database [33] was used to assign total sugars. This table includes 1000 generic/unbranded food items and 9600 branded items (data provided by industry) [33]. Information on total sugar content was missing in 1% of all consumed foods. If these foods were known to be significant sources of total sugars and were reported more than 15 times by survey participants, we assigned them a total sugar value that was based on similar products included in the Swiss Food Composition Database, manufacturer websites, or the online French Food Composition Table (Ciqual) [36]. *Added sugars* were defined as all sugars that are added during food processing and preparation (e.g., white and brown sugar/sucrose, glucose, high-fructose corn syrup, dextrose, fructose, honey, invert sugar, and lactose) [35,37]. Added sugars did not include lactose in dairy products nor sugars naturally occurring in fruit juices and concentrates, and unprocessed foods, such as fruit, vegetables, legumes, potatoes, fish, meat, poultry, and eggs [35,38]. WHO defines *free sugars* as “all sugars that are added during food manufacturing and preparation as well as sugars that are naturally present in honey, syrups, fruit juices and fruit concentrates” [16]. In 2018, Swan et al. [39] suggested a stricter definition of free sugars for the United Kingdom. In addition to the WHO definition, the British authors included all sugars contained in pureed fruit, fruit canned in syrup, and all types of alcoholic and non-alcoholic drinks [39]. We decided to comply with Swan’s definition of free sugars. Appendix A gives an overview of the decisions made and the number of food items assigned to the 30 food groups and their subgroups.

Two dietitians systematically and independently assigned the added sugar content for each food consumed by survey participants. Their estimations were based on standard recipes, information on packaging, or manufacturer websites [37,38]. When information was limited, added sugar content was estimated as the amount of total sugars minus the amount of natural occurring sugars estimated from a similar unsweetened food item (e.g., for apple puree and yogurts, see Appendix A) [37,38]. Free sugars were equal to added sugars, except for the sugars naturally present in all alcoholic and non-alcoholic drinks (including fruit juices and concentrates), pureed fruit, and fruit canned in syrup (see above and Appendix A). 

### 2.4. Covariates 

Sociodemographic information was collected by questionnaire: i.e., sex, age, education (highest degree = university degree: yes/no), nationality (Swiss/non-Swiss). Language region was assigned according to home address. The German-speaking region included the cantons of Aargau, Basel-Land, Basel-Stadt, Bern, Lucerne, St. Gallen, Zurich; the French-speaking region included Geneva, Jura, Neuchatel, and Vaud; and the Italian-speaking region included Ticino. 

### 2.5. Statistical Analyses

The usual daily consumption of total, added, and free sugars as well as total carbohydrate (CHO) and TEI was modeled out of the two 24HDR using the Multiple Source Method (MSM, https://nugo.dife.de/msm) [40]. The latter has been developed to assess the long-term average intake from short-term measurements, accounting for day-to-day variations (within-person variation). In MSM, we assumed that the survey participants were all potential consumers of total, added, and free sugars. The intakes of total, added, and free sugars (in g/day) and the percentages of TEI, CHO, or total sugars are presented by sex, age group, language region, educational level, and nationality. The proportions of persons adhering to international and national recommendations were stratified by the same categories. The main food sources of total added and free sugars are presented within 30 food groups. The average contribution percentage of each group was estimated using two-day mean intake. All the descriptive results were weighted for small deviations in sociodemographic characteristics (age, sex, marital status, place of residency, nationality, and household size), weekday, and seasonality [28]. The weighing strategy tends to limit bias due to the sampling design and non-response; it intends to provide results that are more representative of the Swiss population and of any day of the year. The weighing factors were computed in an iterative process using the 2014 sampling frame as the reference population [28,29]. We carried out all the statistical analyses using STATA version 14 (Stata Corp., College Station, TX, USA).

## 3. Results

Table 1 details the total, added, and free sugar consumption in the Swiss adult population. The mean daily total sugar intake was 107 g (±44), contributing 19% of TEI. The mean daily added sugar intake was 53 g (±36), contributing to 9% of TEI, and the mean daily free sugar intake was 65 g (±40), representing 11% of TEI. Added and free sugar intakes contributed to 48% and 59% of total sugar intake, respectively (Table 1). The proportion of carbohydrate intake consumed as sugars was higher among women than men (49% versus 43% of CHO), whereas a larger proportion of total sugar was consumed as free sugars among men (62% of total sugars, versus 55% in women). Young adults aged 18 to 29 years had a higher mean (±SD) consumption of total (113 g ± 47), added (54 g ± 36), and free sugar (66 g ± 40) compared to older adults aged 65 to 75 years (total sugar: 99 g ± 38; added sugar: 42 g ± 25; free sugar: 53 g ± 28) (Table 1). Individuals living in the Italian-speaking region had a lower mean intake of free sugar (51 g ± 38) compared to the French-speaking (64 g ± 38) and German-speaking regions (66 g ± 40). Intakes of total, added, and free sugar were similar between persons with or without a university degree as well as individuals with or without Swiss nationality (Table 1).

Table 2 shows the main food group sources of total, added, and free sugars. The three main sources of total sugars were (1) sweet products (28%), (2) fruit (21%), and (3) beverages (20%). Main sources of added sugars were (1) sweet products (55%), (2) beverages (17%), and (3) dairy products (10%). Main sources of free sugars were (1) sweet products (47%, in particular sweet spreads (15%) and cakes/cookies (11%)), (2) beverages (29%, mainly fruit and vegetable juices (13%) and soft drinks/SSB (12%)), and (3) dairy products (9%, with yogurt accounting for 6%). These same three food groups were the largest contributors of free sugar, independently of age and gender (Table 2 and Table 3). However, among younger adults, a bigger proportion of free sugars came from beverages, mainly SSBs (20%), compared to older adults, whose main source of free sugar came from sweet products.

Table 4 provides information about adherence to WHO and national recommendations. Respectively, 44% of women and 45% of men had a free sugar intake below 10% of TEI. Among people aged 18 to 29, 30 to 64, and 65 to 75 years, 36%, 45%, and 53% had free sugar intake below 10% of TEI (*p* = 0.001). A bigger proportion of individuals from the Italian-speaking region had a free sugar intake below 10% of TEI compared to the French-speaking and German-speaking regions, which were 62%, 42%, and 44%, respectively (*p* = 0.002). The prevalence of the population with a free sugar intake below 5% of TEI was 8% (Table 4); this figure was slightly higher in the Italian-speaking region, with a proportion of 13%.

## 4. Discussion

In this representative sample of the Swiss population, total sugar intake represented 19% of TEI (mean: 107 g/day ± 44), while added and free sugar intakes represented 9% of TEI (mean: 53 g/day ± 36) and 11% of TEI (65 g/day ± 40), respectively. Adherence to the WHO recommendations for free sugar intake was generally low, with more than half of the population (56%) reporting a free sugar intake above the recommended 10% of TEI, and a large majority (92%) consuming more than 5% of TEI from free sugar. The consumption of total, added, and free sugar was higher among young adults (18–29 years) compared to older adults (65–75 years), whereas individuals living in the Italian-speaking region had a lower mean intake of free sugar compared to the other two linguistic regions (German and French). In Swiss adults, the foods that mostly contributed to free sugar intake were sweet products, non-alcoholic beverages, and dairy products, with SSB contributing the most to free sugar intake amongst young adults.

The current consumption of total, added, and free sugars (respectively 19%, 9%, and 11% of TEI) among Swiss adults is slightly lower than the intakes reported in the Dutch adult population (19 to 69 years old), which were respectively 21%, 12%, and 14% of TEI [38]. A recent review of nationally representative dietary surveys across the world reported that the total sugar intake among adults aged 18 to 65 ranged from 13.5% to 24.6%, with Italian men having the lowest consumption and young American women having the highest [21]. Similarly, the consumption of total sugar in our survey was the lowest among individuals living in the Italian-speaking region of Switzerland. The consumption of added sugar by the Swiss population is consistent with global estimates reported by Newens et al. (2016) showing that added sugar intake ranged from 7.2% in Brazil and Norway to 16.3% in the United States (USA) [21]. Since this survey was the first of its kind in Switzerland, comparisons of the consumption of total, added, and free sugars over time could not be performed. However, worldwide trends in dietary sugar intake in 10 European countries, as well as Australia, New Zealand, and the USA suggest that dietary sugar intake values have been mainly decreasing or stable between 1971–2012, although estimates of mean population dietary sugars intake may be increasing in some specific subpopulations [41].

Consistent with our findings, a recent review of dietary sources of sugars in Europe reported that sweet products, non-alcoholic and alcoholic beverages and dairy products were the foods contributing the most to added sugar intake in both adults and children [20]. In France, sweet products accounted for 60% of total added sugar in women and 61% in men aged 18 to 79 years, whereas they respectively accounted for 53% and 47% in women and men aged 19 to 69 years living in the Netherlands (Switzerland: 57% and 54%) [20]. As for non-alcoholic and alcoholic beverages, they represented 12% of the total added sugar in the diets of French women, while the comparative figures were 16% for French men, 24% for Dutch women, and 31% for Dutch men (Switzerland: 14% and 21%) [20]. The same European review [20] found that SSB alone (excluding fruit juices) accounted for 7% and 10% of total added sugar among French females and males, and 18% and 26% among Dutch females and males, respectively (Switzerland: 10% and 17%). Similarly, SSB and sweet bakery products were the primary contributor to added sugar intake in the USA, followed by candy and other desserts (e.g., ice cream) [22].

A few previous studies have investigated adherence to the free sugar guidelines of the WHO among adults and children [38,42]. The study in adults found that 4% of Dutch men and women aged 19 to 69 reported a free sugar consumption below 5% of TEI, while a third of the women and the men (29% and 33%) observed <10% of TEI [38]. This suggests that the adherence to the WHO guidelines was slightly better in Switzerland, despite the general relative low adherence. These differences might also be partly explained by a lower participation rate in menuCH, increasing the risk of participants being less representative of the general population, i.e., more health-conscious. In addition, although the methods used to estimate sugar consumption in the Netherlands and in Switzerland were similar (i.e., two 24HDR, modeling of usual intake, and nutrition experts to assign added sugar content in foods), we cannot exclude discrepancies.

Two main limitations of the present study are (1) the frequent under-reporting of food items that are high in sugar, and (2) the imprecise estimations of the total, added, and free sugar contents of some food products due to constant changes in the content and ingredient quantity used by manufacturers (food reformulation), and the lack of consensus on added and free sugar definitions. To attenuate this issue, sugar content estimations were performed by two dietitians independently based on standard recipes, information on packaging, or manufacturer websites, and the decisions made were documented in a flow chart, following the example of Sluik et al. in the Netherlands [38]. Since added sugars are more narrowly defined than free sugars and because foods rich in added sugars are the main under-reported food items by both adults and children [21], the percentage of total energy intake coming from added and free sugars is likely to be even higher. Another limitation of this study was that sugar intake among children could not be estimated, since data on Swiss children’s diet have not yet been collected: therefore, the present findings can only be generalized to the Swiss adult population. The main strength of this study lies in the inclusion of two non-consecutive 24HDR conducted by trained dietitians, which enabled the estimation of habitual dietary intake and the removal of some intra-individual variation using the MSM. This study is also the first ever to report data regarding sugar consumption in a relatively representative sample of the Swiss adult population.

## 5. Conclusions

In conclusion, more than half of women and men had free sugar intakes above the updated WHO recommendation of 10% of TEI, and 92% of the population had an intake of free sugar above 5% of TEI. Future surveys should include data about children’s dietary intake, but available data among adults show that those within the Swiss population consume more sugar than they should, and a major public health effort would be necessary to increase the adherence to the recommendations.

## Figures and Tables

**Table 1 nutrients-11-01117-t001:** Weighted habitual intakes of total, added, and free sugars. TEI: total energy intake.

			Total sugars							Added sugars							Free sugar						
	N (%)	Weighted N (%)	Mean in g	SD	Median in g	P25	P75	% TEI (mean)	% Total CHO (mean)	Mean in g	SD	Median in g	P25	P75	% TEI (mean)	% Total sugars (mean)	Mean in g	SD	Median in g	P25	P75	% TEI (mean)	% Total sugars (mean)
**All**	2057 (100)	4,627,878 (100)	107	44	100	76	130	19.3	46.0	53	36	44	29	66	9.3	48.0	65	40	56	38	83	11.5	58.6
**Sex** Men	933 (45)	2,305,141 (50)	114	49	108	80	142	18.1	43.4	61	41	50	33	79	9.4	51.2	73	44	65	43	96	11.4	62.0
Women	1124 (55)	2,322,737 (50)	99	37	94	73	119	20.5	48.5	46	28	39	27	56	9.3	44.8	57	32	50	34	69	11.5	55.3
**Age** 18–29 y	400 (19)	870,489 (19)	113	47	109	79	137	19.6	44.0	63	41	53	33	81	10.5	52.7	75	45	67	42	97	12.7	63.5
30–64 y	1319 (64)	3,108,362 (67)	107	44	100	77	130	19.2	45.9	53	36	44	29	65	9.3	48.0	65	39	56	38	82	11.4	58.6
65–75 y	338 (16)	649,026 (14)	99	38	94	73	121	19.5	48.7	42	25	37	26	52	8.1	41.7	53	28	47	33	66	10.2	52.4
**Linguistic region** German-speaking	1341 (65)	3,183,216 (69)	110	44	104	78	134	19.5	46.2	54	36	46	30	68	9.4	47.6	66	40	58	39	87	11.6	58.2
French-speaking	502 (24)	1,187,738 (26)	103	43	95	73	121	19.1	46.1	52	35	44	29	64	9.4	49.3	64	38	56	36	79	11.6	60.2
Italian-speaking	214 (10)	256,925 (6)	89	44	84	61	108	17.3	41.9	43	35	33	24	50	8.1	46.9	51	38	43	31	60	9.7	56.1
**Education (highest degree) ^1^** No university degree	1057 (51)	2,210,585 (48)	105	46	97	72	127	19.3	46.1	54	38	44	28	66	9.6	48.9	65	42	56	36	82	11.7	59.5
University degree	997 (48)	2,405,018 (52)	108	41	104	79	132	19.3	45.9	52	33	44	31	66	9.1	47.0	64	36	56	39	83	11.2	57.8
**Nationality ^1^** Swiss	1789 (87)	3,470,404 (75)	107	43	101	77	131	19.5	46.2	53	34	45	30	68	9.4	47.9	65	38	58	38	84	11.5	58.7
Non-Swiss	265 (13)	1,145,199 (25)	105	45	97	73	126	18.8	45.1	53	40	41	28	63	9.2	48.0	64	43	53	36	79	11.2	58.1

^1^ Three survey participants did not answer this question.

**Table 2 nutrients-11-01117-t002:** Mean contribution by food group for total, added, and free sugars in the entire population and by sex.

	All (*n* = 2057, Weighted *n* = 4,627,878)			Men (*n* = 933, Weighted *n* = 2,305,141)			Women (*n* = 1,124, Weighted *n* = 2,322,737)		
Food group	Total Sugars (%)	Added Sugars (%)	Free Sugars (%)	Total Sugars (%)	Added Sugars (%)	Free sugars (%)	Total Sugars (%)	Added Sugars (%)	Free Sugars (%)
Tubercles and potato products	0.5	0.4	0.3	0.6	0.4	0.2	0.5	0.4	0.3
Vegetables	6.6	0.9	0.6	6.1	0.9	0.6	7.2	0.8	0.6
Fruit	20.7	0.2	1.3	17.5	0.2	1.0	23.8	0.2	1.6
Nuts, seeds, and olives	0.3	0.0	0.0	0.2	0.0	0.0	0.4	0.0	0.0
**Dairy products**	**13.0**	**10.0**	**8.6**	**12.4**	**9.5**	**8.2**	**13.6**	**10.5**	**8.9**
Milk	5.0	0.1	0.1	4.8	0.1	0.1	5.1	0.1	0.1
Dairy beverages	1.0	1.1	1.0	0.9	0.8	0.7	1.1	1.4	1.2
Dairy desserts	1.1	1.8	1.5	1.1	1.5	1.3	1.2	2.0	1.7
Yogurt	5.5	6.7	5.8	5.3	6.8	6.0	5.6	6.6	5.5
Cottage cheese, quark	0.4	0.4	0.3	0.3	0.3	0.2	0.5	0.5	0.4
Bread, bread products, and dough	2.8	2.0	1.6	2.9	1.3	1.1	2.7	2.6	2.2
Breakfast cereals	1.1	2.7	2.2	1.2	2.5	2.1	1.1	2.9	2.4
Other starchy foods, cereals, and legumes	0.8	0.1	0.1	1.0	0.1	0.0	0.7	0.1	0.1
**Sweet products**	**28.1**	**55.4**	**47.4**	**29.2**	**54.0**	**46.5**	**27.0**	**56.7**	**48.3**
Table sugar	4.2	8.8	7.4	5.1	10.1	8.6	3.4	7.6	6.1
Honey, jam, and other sweet sauces/spreads	8.5	16.6	14.7	9.4	17.3	15.2	7.6	15.9	14.1
Confectionary and candies	3.7	7.3	6.2	3.2	5.9	4.9	4.1	8.7	7.4
Pure chocolate	2.3	5.0	4.1	2.1	3.9	3.3	2.5	6.0	4.9
Syrups	1.0	1.8	1.6	1.0	1.7	1.5	1.1	1.9	1.7
Ice cream	1.2	2.5	2.1	1.2	2.3	1.9	1.3	2.6	2.3
Cakes and cookies	7.1	13.4	11.3	7.2	12.8	10.9	7.0	14.0	11.7
**Beverages**	**19.9**	**17.4**	**29.4**	**22.8**	**20.5**	**32.4**	**17.1**	**14.2**	**26.4**
Fruit and vegetable juices	8.3	1.8	12.9	8.3	1.8	12.4	8.2	1.8	13.5
Soft drinks	9.1	13.2	12.1	11.8	16.6	15.3	6.4	9.9	8.9
Water	0.1	0.2	0.2	0.1	0.2	0.2	0.1	0.3	0.2
Coffee and tea	0.8	0.9	0.8	0.7	0.8	0.7	0.9	1.0	0.8
Alcoholic drinks	1.6	1.2	3.4	1.8	1.1	3.9	1.4	1.3	2.9
Processed meat	0.4	2.1	1.4	0.5	2.5	1.5	0.3	1.7	1.2
Processed fish and seafood	0.0	0.1	0.1	0.0	0.2	0.2	0.0	0.1	0.0
Sauces, condiments, and flavorings	2.7	6.1	4.6	2.9	5.6	4.3	2.6	6.6	4.9
Soups	1.1	0.7	0.5	0.9	0.6	0.5	1.3	0.8	0.6
Salty snacks	0.3	0.5	0.4	0.3	0.4	0.3	0.3	0.6	0.4
Others/miscellaneous	1.5	1.1	1.3	1.5	1.0	1.1	1.5	1.1	1.5

**Table 3 nutrients-11-01117-t003:** Mean contribution by food group for total, added, and free sugars by age group.

	18 to 29 years old (*n* = 400, Weighted *n* = 870,489)			30 to 64 years old (*n* = 1319, Weighted *n* =3,108,362)			65 to 75 years old (*n* = 338, Weighted *n* = 649,026)		
Food group	Total Sugars (%)	Added Sugars (%)	Free Sugars (%)	Total Sugars (%)	Added Sugars (%)	Free Sugars (%)	Total Sugars (%)	Added Sugars (%)	Free Sugars (%)
Tubercles and potato products	0.7	0.5	0.3	0.5	0.3	0.3	0.5	0.4	0.3
Vegetables	6.1	1.2	1.0	6.8	0.8	0.6	6.6	0.4	0.2
Fruit	15.3	0.2	0.8	20.7	0.2	1.3	27.7	0.3	1.8
Nuts, seeds, and olives	0.3	0.0	0.0	0.3	0.0	0.0	0.2	0.0	0.0
**Dairy products**	**12.6**	**8.7**	**7.5**	**13.1**	**10.2**	**8.7**	**12.8**	**10.9**	**9.4**
Milk	5.6	0.2	0.1	4.9	0.0	0.0	4.5	0.0	0.0
Dairy beverages	1.2	0.8	0.8	1.0	1.3	1.1	0.6	0.6	0.5
Dairy desserts	1.0	1.4	1.1	1.2	1.8	1.6	1.2	1.9	1.6
Yogurt	4.3	5.8	5.1	5.6	6.7	5.7	6.2	8.0	7.0
Cottage cheese, quark	0.6	0.5	0.3	0.4	0.4	0.3	0.4	0.3	0.3
Bread, bread products, and dough	2.9	1.6	1.2	2.9	2.0	1.7	2.4	2.4	1.8
Breakfast cereals	2.0	4.4	3.6	1.0	2.5	2.0	0.7	1.5	1.4
Other starchy foods, cereals, and legumes	1.2	0.1	0.1	0.7	0.1	0.1	0.7	0.1	0.1
**Sweet products**	**24.2**	**43.8**	**37.6**	**28.6**	**56.2**	**48.0**	**30.8**	**67.1**	**57.8**
Table sugar	3.2	6.1	5.2	4.6	9.6	8.1	3.7	8.6	7.1
Honey, jam, and other sweet sauces/spreads	6.4	11.8	10.4	8.0	15.5	13.5	13.4	28.4	25.8
Confectionary and candies	4.9	9.0	7.8	3.7	7.3	6.2	2.2	5.2	4.3
Pure chocolate	1.8	3.5	3.0	2.5	5.5	4.5	1.7	4.3	3.5
Syrups	1.0	1.6	1.4	1.1	1.8	1.6	1.0	1.7	1.6
Ice cream	1.2	2.1	1.8	1.2	2.3	2.0	1.5	3.7	3.2
Cakes and cookies	5.7	9.6	8.0	7.5	14.1	12.0	7.3	15.4	12.4
**Beverages**	**28.0**	**27.4**	**37.7**	**19.3**	**16.6**	**28.8**	**12.5**	**7.5**	**21.0**
Fruit and vegetable juices	9.3	2.6	13.4	8.1	1.7	12.8	7.6	1.3	13.1
Soft drinks	15.9	22.1	20.1	8.5	12.5	11.4	3.0	4.8	4.3
Water	0.1	0.2	0.2	0.1	0.3	0.2	0.0	0.1	0.1
Coffee and tea	0.9	1.1	1.0	0.9	0.9	0.8	0.5	0.5	0.3
Alcoholic drinks	1.8	1.3	3.0	1.6	1.2	3.6	1.4	0.9	3.2
Processed meat	0.5	2.2	1.8	0.4	2.1	1.3	0.4	1.8	1.1
Processed fish and seafood	0.0	0.2	0.2	0.0	0.1	0.1	0.0	0.0	0.0
Sauces, condiments, and flavorings	3.4	6.8	5.6	2.8	6.1	4.6	1.8	5.3	3.4
Soups	0.9	0.8	0.5	1.1	0.8	0.6	1.2	0.5	0.3
Salty snacks	0.4	0.5	0.3	0.3	0.5	0.3	0.3	0.8	0.5
Others/miscellaneous	1.5	1.6	1.8	1.5	1.0	1.2	1.5	0.8	0.8

**Table 4 nutrients-11-01117-t004:** Adherence to international and national guidelines.

	*n* (%)	Weighted *n* (%)	Free sugar: <10% of TEI (%) ^2,3^	*p*-Value ^4^	Free Sugar: <5% of TEI (%) ^5^	*p*-Value ^4^	Free sugar: max. 50 g/day ^3^	*p*-Value ^4^
**All**	2057 (100)	4,627,878 (100)	44%		8%		42%	
**Sex** Men	933 (45)	2,305,141 (50)	45%		9%		35%	
Women	1124 (55)	2,322,737 (50)	44%	0.65	8%	0.32	50%	**<0.001**
**Age** 18–29 y	400 (19)	870,489 (19)	36%		7%		34%	
30–64 y	1319 (64)	3,108,362 (67)	45%		8%		42%	
65–75 y	338 (16)	649,026 (14)	53%	**<0.001**	11%	0.48	54%	**<0.001**
**Linguistic region** German-speaking	1341 (65)	3,183,216 (69)	44%		9%		40%	
French-speaking	502 (24)	1,187,738 (26)	42%		6%		42%	
Italian-speaking	214 (10)	256,925 (6)	62%	**0.002**	13%	**0.039**	64%	**<0.001**
**Education (highest degree) ^1^** No university degree	1057 (51)	2,210,585 (48)	43%		8%		43%	
University degree	997 (48)	2,405,018 (52)	46%	0.44	9%	0.97	41%	0.46
**Nationality** ^1^ Swiss	1789 (87)	3,470,404 (75)	43%		8%		42%	
Non-Swiss	265 (13)	1,145,199 (25)	50%	0.09	10%	0.24	43%	0.68

^1^ Three survey participants did not answer this question. ^2^ Strong recommendation from World Health Organization. ^3^ National recommendation from the Federal Commission on Nutrition. ^4^ Differences between groups were estimated using chi-squared tests for weighted data. ^5^ Conditional recommendation from World Health Organization.

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
