# Peer review of "Total, Added, and Free Sugar Consumption and Adherence to Guidelines in Switzerland: Results from the First National Nutrition Survey menuCH"

_nutrients, 2019, doi:10.3390/nu11051117_

Round 1
Reviewer 1 Report
This is an excellent paper with interesting data well presented. There is much interest in Free Sugars intakes which have not been well-researched in the past. This paper adds some important data to this field. The supplementary data detailing decisions made during the production of the data will be useful for other researchers wishing to work on free sugars intake.
I have very few comments to make:
I was unhappy with the use of the word ‘sweets’ lines 25, 159, 165 and Tables 2a and 2b. I think ‘sweet products’ or ‘sweet foods’ would be better.
Similarly, I was unhappy with the use of the word ‘dairies’ lines 161, 194. I think ‘dairy products’ or ‘sweet dairy products’ would be better.
Line 247 -----children’s----- or -----childhood-----
I found the supplementary table very interesting but confusing in places. I suggest
In the title:
------- added sugars from total sugars, free sugars are assumed to be the same as added sugars except where marked * in which case 100% of total sugars is assumed to be free sugars. These decisions were largely based---------
In my experience your estimate of added and free sugars from Breakfast cereals seems rather low as many are sweetened with sugar.
In the foot note, I suggest:
Replace ‘decomposed’ with ‘disaggregated’
Author Response
Reviewer #1
Point 1: This is an excellent paper with interesting data well presented. There is much interest in Free Sugars intakes which have not been well-researched in the past. This paper adds some important data to this field. The supplementary data detailing decisions made during the production of the data will be useful for other researchers wishing to work on free sugars intake.
Answer 1: We thank you for your thorough, constructive and supportive review. Please find below a point-by-point response to your comments.
Point 2: I have very few comments to make: I was unhappy with the use of the word ‘sweets’ lines 25, 159, 165 and Tables 2a and 2b. I think ‘sweet products’ or ‘sweet foods’ would be better. Similarly, I was unhappy with the use of the word ‘dairies’ lines 161, 194. I think ‘dairy products’ or ‘sweet dairy products’ would be better.
Answer 2: We fully agree on the comments. The word ‘sweets’ has been replaced by ‘sweet products’ throughout the manuscript (lines 25, 162, 168, 197, and Tables 2a/b). Similarly, the term ‘dairies’ has been replaced by ‘dairy products’ (lines 165, 197).
Point 3: Line 247 -----children’s----- or -----childhood-----
Answer 3: Changed (line 254)
Point 4: I found the supplementary table very interesting but confusing in places. I suggest in the title: ------- added sugars from total sugars, free sugars are assumed to be the same as added sugars except where marked * in which case 100% of total sugars is assumed to be free sugars. These decisions were largely based---------
Answer 4: We thank you for your relevant suggestion and have changed the table’s title accordingly: ‘Table S1: Decision flow-chart of estimated percentage for added sugars from total sugars. Free sugars are assumed to be the same as added sugars except where marked * in which case 100% of total sugars is assumed to be free sugars. These decisions were largely based upon the definition by Sluik et al. in the Netherlands, Bowman in the U.S., and Swan et al. in the UK. ‘. We also added two supplementary figures (S1 and S2) for a better overview of the definitions of total, free and added sugars.
Point 5: In my experience your estimate of added and free sugars from Breakfast cereals seems rather low as many are sweetened with sugar.
Answer 5: We fully acknowledge your point. This is indeed difficult to estimate the proportion of added sugar in branded breakfast cereals, especially with the proportion of each ingredient is absent in the ingredient list. In Switzerland, there is, however, a long tradition on muesli mixes, which contain a relatively high proportion of natural nuts, seeds and dried fruits (e.g. sultanas, apples, etc.). According to the Swiss Food Composition Database, unsweetened dried fruits naturally contain about 60-70g of total sugars per 100g, and nuts/seeds about 5g/100g. This explains why we estimated a relatively low proportion of added sugar from total sugar in branded breakfast cereals (75-90%, often closer to 90% than 75%).
Point 6: In the foot note, I suggest: Replace ‘decomposed’ with ‘disaggregated’
Answer 6: Changed (Supplementary Table S1).
Reviewer 2 Report
A useful paper that is very well written and provides valuable background data on population intakes of these 3 carbohydrate types. Some minor edits/suggestions:
1) could the authors provide a little more description (examples) of how they derived the food groups? are these based on other published food groups, a purely pragmatic approach or published elsewhere?
2) I think the authors would benefit from spending more time explaning their methods for allocation sugar as total, added or free. Supplementary table is not completely clear. I think a flow chart might help readers unfamiliar with the subtle differences between the various definitions of the different sugar form.
3) could the authors comment in their discussions about the different definitions used and what that means for interpreting results (even for free sugars alone).
4) please comment on how recent the composition data was - i.e. most likely to reflect reformulation
5. Could the authors consider looking at intakes by body weight status (i.e normal, overweight/obese?)
Author Response
Reviewer #2
Point 1: A useful paper that is very well written and provides valuable background data on population intakes of these 3 carbohydrate types.
Answer 1: We thank you for your thorough, constructive and supportive review. Please find below a point-by-point response to your comments.
Point 2: Some minor edits/suggestions:1) could the authors provide a little more description (examples) of how they derived the food groups? are these based on other published food groups, a purely pragmatic approach or published elsewhere?
Answer 2: We classified all foods and beverages into 30 food groups, based on their original classification from GloboDiet® (18 groups and 85 subgroups) and on their nutritional similarity regarding sugar content. For instance, cheese and fresh meat were not presented as separate groups as not main providers of sugars. GloboDiet®, previously known as EPIC-Soft®, is a software/program that guide dietitians in their 24-hour dietary recalls. Because this software has been used in most national nutrition surveys in Europe, we used its classification to allow comparison at food group level between European countries. The article ‘Total, Free, and Added Sugar Consumption and Adherence to Guidelines: The Dutch National Food Consumption Survey 2007–2010’ by Sluik et al. in the Netherlands also uses the same food group classification given by GloboDiet® (17 groups and 85 subgroups, reference 38). We slightly amended the manuscript (see lines 92-94), but think Supplementary Table S1 provides enough information regarding foods included in each food group.
Point 3: 2) I think the authors would benefit from spending more time explaining their methods for allocation sugar as total, added or free. Supplementary table is not completely clear. I think a flow chart might help readers unfamiliar with the subtle differences between the various definitions of the different sugar form.
many are sweetened with sugar.
Answer 3: We fully acknowledge your comment. We added two Supplementary Figures (S1 and S2) to assist readers with definitions and calculation of total, added and free sugars (lines 96-97).
Point 4: 3) could the authors comment in their discussions about the different definitions used and what that means for interpreting results (even for free sugars alone).
Answer 4: We agree on your point and amended the manuscript accordingly (lines 236-245):
‘Two main limitations of the present study are 1) the frequent under-reporting of food items high in sugar, and 2) the imprecise estimations of total, added and free sugars contents of some food products, due to constant changes in content and ingredient quantity used by manufacturers (food reformulation), and the lack of consensus on added and free sugar definitions. To attenuate this issue, sugar content estimations were performed by two dietitians independently based on standard recipes, information on packaging or manufacturer websites, and decisions made were documented in a flow-chart, following the example of Sluik et al. in the Netherlands. Because added sugars are more narrowly defined than free sugars and because foods rich in added sugars are the main under-reported food items by both adults and children [41], the percentage of total energy intake coming from added and free sugars is likely to be even higher.'
Point 5: 4) please comment on how recent the composition data was - i.e. most likely to reflect reformulation
Answer 5: The food consumption data were linked to an extended research version of the 2015 Swiss Food Composition Database by the Swiss Federal Food Safety and Veterinary Office, using the software FoodCASE. The manuscript has been amended as follows:
‘According to the description reported by the dietitians, each food item was linked to the best match in an extended research version of the 2015 Swiss Food Composition Database [33] using FoodCASE (Premotec GmbH, Winterthur, Switzerland) [34].’ (line 91 + Answer 5).
Point 6: 5) Could the authors consider looking at intakes by body weight status (i.e., normal, overweight/obese?)
Answer 6: We think this aspect is important but needs caveats in the interpretation of the results due to probable higher underreporting in overweight/obese population and the cross-sectional design of the survey. For this reasons, we plan to discuss the association between sugar intake and obesity in another manuscript.